# Matrix Metalloproteinase-2 Inhibition in Acute Ischemia-Reperfusion Heart Injury—Cardioprotective Properties of Carvedilol

**DOI:** 10.3390/ph14121276

**Published:** 2021-12-07

**Authors:** Monika Skrzypiec-Spring, Joanna Urbaniak, Agnieszka Sapa-Wojciechowska, Jadwiga Pietkiewicz, Alina Orda, Bożena Karolko, Regina Danielewicz, Iwona Bil-Lula, Mieczysław Woźniak, Richard Schulz, Adam Szeląg

**Affiliations:** 1Department of Pharmacology, Wrocław Medical University, 50-345 Wrocław, Poland; adam.szelag@umed.wroc.pl; 2Lower Silesian Oncology Centre, 53-413 Wrocław, Poland; urbaniak.joanna@dco.com.pl; 3Department of Clinical Chemistry, Wrocław Medical University, 50-556 Wrocław, Poland; agnieszka.sapa-wojciechowska@umw.edu.pl (A.S.-W.); iwona.bil-lula@umw.edu.pl (I.B.-L.); mieczyslaw.wozniak@umed.wroc.pl (M.W.); 4Department of Biochemistry, Wrocław Medical University, 50-368 Wrocław, Poland; jadwiga.pietkiewicz@umed.wroc.pl (J.P.); regina.danielewicz@umed.wroc.pl (R.D.); 5Department of Cardiology, Wrocław Medical University, 50-556 Wrocław, Poland; alina.orda@op.pl (A.O.); bozena.karolko@umed.wroc.pl (B.K.); 6Departments of Pediatrics and Pharmacology, University of Alberta, Edmonton, AB T6G 2S2, Canada; richard.schulz@ualberta.ca

**Keywords:** β-blockers, carvedilol, matrix metalloproteinase-2, ischemia-reperfusion injury, isolated heart perfusion

## Abstract

Matrix metalloproteinase 2 (MMP-2) is activated in hearts upon ischemia-reperfusion (IR) injury and cleaves sarcomeric proteins. It was shown that carvedilol and nebivolol reduced the activity of different MMPs. Hence, we hypothesized that they could reduce MMPs activation in myocytes, and therefore, protect against cardiac contractile dysfunction related with IR injury. Isolated rat hearts were subjected to either control aerobic perfusion or IR injury: 25 min of aerobic perfusion, followed by 20 min global, no-flow ischemia, and reperfusion for 30 min. The effects of carvedilol, nebivolol, or metoprolol were evaluated in hearts subjected to IR injury. Cardiac mechanical function and MMP-2 activity in the heart homogenates and coronary effluent were assessed along with troponin I content in the former. Only carvedilol improved the recovery of mechanical function at the end of reperfusion compared to IR injury hearts. IR injury induced the activation and release of MMP-2 into the coronary effluent during reperfusion. MMP-2 activity in the coronary effluent increased in the IR injury group and this was prevented by carvedilol. Troponin I levels decreased by 73% in IR hearts and this was abolished by carvedilol. Conclusions: These data suggest that the cardioprotective effect of carvedilol in myocardial IR injury may be mediated by inhibiting MMP-2 activation.

## 1. Introduction

β-blockers improve the oxygen supply/demand ratio by reducing myocardial oxygen consumption via a decrease in heart rate and myocardial contractility [1]. Several clinical studies showed the ability of β-blockers to decrease the incidence of coronary heart disease and mortality in patients with coronary heart disease and acute coronary syndromes [2,3]. The first generation β-blockers are represented by propranolol, sotalol, oxprenolol and nadolol, the second by metoprolol, atenolol, and bisoprolol, and the third by carvedilol, nebivolol, and labetalol [3]. The cardioprotective effects of different β-blockers were tested on animal models. It was shown that carvedilol and nebivolol possess superior cardioprotective efficacy as compared to other β-blockers which failed to protect the myocardium against IR injury [4,5,6,7,8,9]. These observations suggest that the protective activity of some β-blockers is most probably unrelated to β-adrenoceptor inhibition but to other‘-ancillary’- properties.

Carvedilol (CAR) is a third generation, non-selective β-blocker which also possesses α-blocker and anti-oxidant properties. It may prevent oxidative stress-induced activation of transcription factors associated with inflammatory and remodeling processes and inhibit the direct cytotoxicity of reactive oxygen and nitrogen species [10]. It was shown to inhibit MMP-8 in coxackie virus-induced myocarditis and MMP-2 and MMP-9 expression in experimental atherosclerosis [11,12]. As we previously described, CAR reduces the contractile dysfunction of heart muscle by attenuation of MMP-2 activity and decreased degradation of troponin and myofilaments in hearts subjected to experimental autoimmune myocarditis [13]. Nebivolol (NEB) is a third generation, highly selective β1-adrenoceptor antagonist endowed with the ability to induce nitric oxide release from the endothelium [14]. Moreover, it was shown to attenuate MMP-2 and MMP-9 activities in experimental renovascular hypertension and renal IR injury [15,16]. In contrast, metoprolol (MET) represents a second generation, selective β1-adrenoceptor antagonist with no pleiotropic actions disclosed.

IR injury occurs as a result of acute oxidative stress following reperfusion after ischemia. It can be described as a cascade of pathophysiological events including the biosynthesis of reactive oxygen and nitrogen species and cytokines and activation of MMPs, in particular MMP-2 [17,18,19]. Although MMPs are primarily known for their ability to cleave substrates in the extracellular matrix, it was also shown that the acute contractile dysfunction in IR injury is also caused by the activation of intracellularly localized MMP-2 which results in troponin I, myosin light chain-1, α-actinin and titin degradation [20,21].

The properties of CAR and NEB to inhibit MMPs, and thus, prevent the degradation of sarcomeric proteins, such as troponin I may provide the underlying pharmacologic rationale for the use of these drugs as a first choice for acute coronary syndromes. The present study was, therefore, performed to test the hypothesis that the cardioprotective action of CAR or NEB in the setting of IR injury is related to their ability to inhibit MMPs activity and troponin I degradation in heart tissue.

## 2. Results

### 2.1. Carvedilol Protects Hearts from Ischemia-Reperfusion Injury

Stable mechanical function of hearts perfused for 75 min in aerobic conditions was observed (data not shown). IR injury caused a significant reduction in the heart rate and mechanical function recovery expressed as LVDP and RPP in the reperfusion period (Figure 1a–c). CAR 0.1 μM significantly improved the recovery of mechanical function at the end of the reperfusion period in comparison to the control IR injury group (*p* < 0.01; Figure 1a). The higher CAR concentrations, as well NEB and MET in all tested concentrations, did not significantly improve the recovery after IR injury (Figure 1a–c).

### 2.2. Carvedilol Influences MMP-2 Activity

The differences when comparing MMP-2 activities in the coronary effluent between carvedilol treated groups were not significant (Figure 2a). Additionally, at higher CAR concentrations, as well as with NEB and MET at all tested concentrations, there were no significant differences between groups (data not shown).

Nevertheless, a significant increase of MMP-2 activity in the coronary effluent, calculated as the ratio of 45 and 25 min perfusion time values was seen in IR injury group, as well as CAR 1 and CAE 10 groups in comparison with the C group (*p* < 0.0001; Figure 2c–e). This increase in MMP-2 activity was significantly lower in the IR injury with CAR 0.1 μM group (Figure 2c). NEB and MET in all tested concentrations had no significant influence on the increase of MMP-2 activity evoked by IR which was significantly higher than in the C group (*p* < 0.0001, Figure 2d,e).

MMP-2 activity in the heart tissue was assessed only at the end of the experiment showing no significant changes between groups (data not shown).

### 2.3. Carvedilol Has No Effect on MMP-2 Activity In Vitro

Carvedilol did not inhibit the activity of MMP-2 when run out on gel zymograms incubated with 0.1 μM carvedilol (Figure 3a,b).

### 2.4. Carvedilol Does Not Change MMP-2 mRNA Expression in Hearts Subjected to Ischemia-Reperfusion

Real time PCR revealed no significant changes in MMP-2 mRNA expression between groups (Figure 4).

### 2.5. Carvedilol Does Not Affect MMP-2 Content in Coronary Effluent

IR injury caused a significant increase in MMP-2 content, assessed by western blot, in coronary effluent in the second minute of reperfusion in both IR and IR-CAR 0.1 groups, but there were no significant differences between IR and IR-CAR 0.1 groups (Figure 5a,b).

There were no significant differences in MMP-2 content in heart tissue between groups (data not shown).

### 2.6. Carvedilol Abolishes Troponin I Level in Heart Tissue

Analysis of troponin I levels in hearts at the end of perfusion showed that IR injury caused a 73% decrease of the level of 31 kDa troponin I which was abolished in the IR-CAR 0.1 μM group (*p* < 0.05; Figure 6a,b). This was not observed NEB and MET in all tested concentrations (Figure 6c,d).

## 3. Discussion

This study demonstrates for the first time that the cardioprotective role of carvedilol (CAR) in experimental IR injury may be caused in part by its ability to inhibit MMP-2 activation. These findings are of clinical importance and suggest a possible beneficial role of CAR over other β-blockers in preventing contractile dysfunction as a consequence of IR injury.

Although MMPs were first recognized for their ability to cleave substrates in the extracellular matrix, more recent studies revealed that hearts undergoing oxidative stress have increased MMP-2 activity. Moreover, it also acts intracellularly, leading to myocardial contractile dysfunction by degradation of α-actinin, myosin light chain-1, titin and troponin I [20,21]. Observations regarding MMPs function in animal hearts with induced IR injury have been verified in human studies. Lalu et al. reported an increase of both MMP-2 and MMP-9 activities in human hearts following cardiopulmonary bypass [22]. It was shown that during cardiopulmonary bypass for coronary artery graft surgery, right atrial biopsies obtained within 10 min after aortic cross-clamp release had increased activities of both MMP-2 and MMP-9 which was inversely correlated with the decrease in contractile function measured 3 h after cross-clamp release [22].

CAR and NEB were both shown to inhibit the activity of various MMPs in conditions of enhanced oxidative stress related to atherosclerosis, myocarditis, periodontitis, hypertension, or renal IR injury [11,12,13,15,16]. However, there were no data on the effect of these drugs on MMP activation in acute IR injury of the heart.

In our study, we showed an improvement in the recovery of cardiac contractile function in IR injury hearts along with a reduction in MMP-2 release into the coronary effluent in 0.1 μM CAR treated hearts. Interestingly, only this lowest concentration of CAR improved the recovery of mechanical function during reperfusion and attenuated the IR injury-induced MMP-2 release into the coronary effluent. The superior role of this low concentration of carvedilol in reducing MMPs activation over higher concentrations was shown in other studies. Jaggi et al. showed that only CAR- 0.1 μM attenuated IR injury in isolated rat hearts by preventing mast cell degranulation [23]. Similarly, a low concentration of CAR (0.05 μM) was shown to exert cardioprotective effects, along with reduction of creatine kinase release, during hypoxia in isolated rat hearts [24]. Our current results are also consistent with our previous findings showing that the lowest concentration of CAR was the most effective in inhibiting MMP-2 activity during acute autoimmune myocarditis [13].

NEB is also known for its anti-MMPs properties [15,16]. However, surprisingly, we did not observe cardioprotective effects and MMP inhibition with NEB despite its equivalent potency at β-receptors to CAR in the concentrations used in the experiment. The explanation of this phenomenon is not straightforward and requires further investigation. NEB is a third generation, highly selective β1-adrenoceptor antagonist endowed with the ability to induce NO release from the endothelium [14]. The protective role of NO in ischemic hearts is exerted by various mechanisms [25,26,27]. Nevertheless, a negative effect of NO on cardiomyocytes through the formation of peroxynitrite was when present at elevated levels also in the presence of superoxide. Mori et al. found that intracoronary administration of the precursor of nitric oxide L-arginine aggravated myocardial stunning in dogs via the biosynthesis of peroxynitrite [28]. A similar effect was achieved using an in vivo canine model with NO donor S-nitroso-N-acetylpenicillamine while NOS inhibition exerted the opposite effect [29,30]. Therefore, the lack of inhibitory action of NEB on MMPs in IR injury of the heart may be, at least in part, explained by overproduction of NO by NEB.

MMPs cause acute cardiac mechanical dysfunction by cleaving sarcomeric proteins, such as troponin I [20]. Furthermore, impaired cardiac contractile function and reduced levels of troponin I were shown in cardiomyocytes of transgenic mice with cardiac specific expression of active MMP-2 [31]. Therefore, inhibition of MMP activity should reduce troponin I degradation. In our study, we showed that in homogenates prepared from IR injury hearts that the troponin I levels were the highest in the IR-CAR 0.1 μM group. These results are in line with previous data showing the effect of CAR on troponin I in patients with adriamycin-induced cardiotoxicity [32] and our previous findings regarding the protective role of CAR on the level of troponin in acute myocarditis along with MMP-2 activity inhibition [13].

The main limitation of our study is that, based on our results, we cannot directly say what is the exact mechanism responsible for the decrease in MMP-2 activity by carvedilol. Based on the results of in vitro experiment we can exclude the direct effect of carvedilol on MMP-2 activity. We can also exclude changes at the transcriptional level as we did not observe differences in MMP-2 mRNA expression between groups, indicating that carvedilol regulates MMPs at the post-transcriptional level. Western blot analysis of MMP-2 content in coronary effluent revealed that there were no changes between the IR and IR-CAR 0.1 μM group while the increase in MMP-2 activity assessed in zymography was significantly lower in the IR injury with CAR 0.1 μM group than in the IR group. Moreover, a decrease in MMP-2 activity in the effluent from hearts treated with carvedilol was accompanied by a decrease in tissue troponin I degradation. These results indicate that the changes of MMP-2 activity resulted from its activation in heart tissue and subsequent release into the coronary effluent in the settings of IR but not from changes in enzyme localization. Further research is needed to explain the exact mechanism by which carvedilol inhibits the activity of MMP-2.

Another limitation of our study is that we did not observe significant changes in MMP-2 tissue activity. However, the tissue MMP-2 activity was assessed only at 30 min of reperfusion, but not within the first two minutes. As a result of reperfusion injury following myocardial ischemia, we showed a rapid and enhanced release of MMP-2 into the coronary effluent which peaked within the first 2 min of reperfusions shown by [19]. As the increased release of activated MMP-2 results from its intracellular activation, consequently, MMP-2 tissue activity at the beginning of reperfusion should be increased. The lack of increased MMP-2 activity in heart homogenates at the end of reperfusion is due to its release from isolated hearts as shown by Cheung et al. [33]. Again, troponin I levels in heart tissue may serve as indirect evidence of MMP activation [19]. Despite this limitation, we suggest that the effects of carvedilol on postischemic cardiac contractility and troponin levels we observe may be due to its ability to inhibit MMP-2 activation and subsequent release.

## 4. Materials and Methods

### 4.1. Animals

All β-blockers (CAR, NEB, MET) were purchased from Sigma-Aldrich (Poznan, Poland). Hearts were obtained from male Wistar rats, weighing 250–350 g, purchased from the Laboratory Animal Center, Wroclaw Medical University.

### 4.2. Heart Perfusion Protocol

Rats were anesthetized with thiopental (75 mg/kg) given intraperitoneally. After sternotomy, hearts were excised and immersed with ice-cold Krebs–Henseleit solution (0.5 mmol/L EDTA, 1.2 mmol/L KH_2_PO_4_, 1.2 mmol/L MgSO4, 3 mmol/L CaCl_2_, 4.7 mmol/L KCl, 11 mmol/L glucose, 25 mmol/L NaHCO_3_, and 118 mmol/L NaCl), gassed with a mixture of 95% O_2_ and 5% CO_2_ in pH 7.4. Then the aorta was cannulated and coronary perfusion with Krebs–Henseleit solution was initiated. Perfusion solution was delivered at constant pressure (60 mmHg) and temperature (37 °C). Left ventricular pressure was monitored by fluid filled latex balloon inserted into the left ventricle. Heart rate was also monitored during the experiments. Left ventricular developed pressure (LVDP) was calculated as the difference between systolic and diastolic pressure of the left ventricular. Cardiac mechanical function was expressed as the heart rate-pressure product (RPP) (calculated as the product of the spontaneous heart rate and LVDP).

The hearts were aerobically perfused for 25 min, then for 20 min subjected to global, no-flow normothermic ischemia and finally aerobically reperfused for 30 min (Figure 7). Control hearts were perfused for 75 min in aerobic conditions. The β-blockers (CAR, NEB, or MET) were infused into the hearts 10 min prior to the onset of ischemia and for the first 10 min of reperfusion, or for 40 min of the control aerobic perfusion (from 15 to 55 min of perfusion). The heart perfusion apparatus had separate perfusion solution reservoirs and a switching valve which allowed the selection between solutions with or without added drugs. MET was dissolved directly into Krebs–Henseleit solution. CAR and NEB were dissolved in dimethyl sulfoxide (DMSO) and subsequently diluted at the desired concentrations with Krebs–Henseleit solution. The concentration of DMSO reaching the heart was < 0.5% (*v*:*v*). Samples of coronary effluent for analysis were collected for an equal time of 2 min immediately prior to ischemia and after 2 and 30 min after reperfusion. At the end of the protocol, hearts were freeze clamped in liquid nitrogen and stored at −80 °C for subsequent biochemical analysis.

### 4.3. Experimental Groups

The hearts were randomly divided into the following 20 experimental groups (*n* = 6 hearts per group) in aerobic or anaerobic (IR) conditions either with no drug or with three different concentrations of tested drugs: CAR (0.1, 1, 10 μM), NEB (0.005, 0.05, 0.5 nM) and MET (0.01, 0.1, 1 μM).

### 4.4. Preparation of Heart Extracts and Concentration of Coronary Effluent

Frozen hearts were crushed into a powder with a mortar and pestle at liquid nitrogen temperature and stored at −80 °C. Prior to the biochemical analysis, the tissue powder was homogenized in 50 mmol/L Tris-HCl (pH 7.4) containing 1 mmol/L dithiothreitol, 3.1 mmol/L sucrose, 2 µg/mL aprotinin, 10 µg/mL soybean trypsin inhibitor, 10 µg/mL leupeptin, and 0.1% Triton X-100. The samples of coronary effluent were concentrated at 4 °C 30-fold using Centricon-10 concentrating vessels purchased from Merck Millipore (Poznan, Poland). The supernatant was collected and stored at −80 °C for analysis. Protein content in both effluent and homogenates was analyzed using Bradford Protein Assay (Bio-Rad, Warszawa, Poland) and bovine serum albumin used as a protein standard.

### 4.5. Measurement of MMP-2 by Gelatin Zymography

MMP-2 activity was assessed in both heart extracts and concentrated coronary effluent. Equal total protein samples were applied to 8% SDS-PAGE gels. Following electrophoresis, gels were rinsed thrice for 20 min each in 2.5% Triton X-100 and afterward twice for 20 min each in incubation buffer (5 mmol/L CaCl_2_, 50 mmol/L Tris-HCl, 150 mmol/L NaCl, and 0.05% NaN3) at room temperature. Subsequently, they were placed for 10 h at 37 °C in incubation buffer and finally were stained in 2% Coomassie Brilliant Blue G, 25% methanol, and 10% acetic acid for 2 h and destained in 30% methanol/ 10% acetic acid solution. Quantification of reactions was performed with the usage of a GS-800 Calibrated Densitometer with Quantity One v.4.6.9 software (BioRad) and the relative MMPs activity was determined and expressed in arbitrary units (AU) calculated on the basis of recombined MMP standard activity. In vitro inhibition of MMPs activity by carvedilol was evaluated by developing zymograms performed on selected heart extracts with the addition of carvedilol to the incubation buffer in comparison to control gels (with no drug added), which served as a model of 100% of MMP-2 activity.

### 4.6. Measurement of Troponin I and MMP-2 by Western Blot

Troponin I content was determined by Western blotting. 30 µg of protein obtained from heart extracts was applied to 15% SDS-PAGE gels. Following electrophoresis, (at 150 V, 20 °C) samples were electroblotted onto a polyvinylidene difluoride membrane (by semi-dry technique; at 25 V, 30 min). A primary monoclonal mouse antibody against cardiac troponin I as well as secondary goat-anti-mouse conjugated with horseradish peroxidase (HRP) were both used at 1:1000 dilution (Thermo Fisher Scientific, Waltham, Massachusetts, USA; BioRad, respectively). The blot was developed using a chemiluminescence assay (ClarityTM Western ECL Substrate, Biorad). Membranes were scanned using ChemiDocTM XRS+ System with Image LabTM Software v.5.2 for data analysis. Rat cardiac troponin I was used for standard curve preparation (Advanced ImmunoChemical Inc., Long Beach, California, USA). MMP-2 content was determined in coronary effluents. 30 μL of each concentrated coronary effluent was applied to 10% SDS-PAGE gels at reducing conditions. After electrophoresis samples were electroblotted for 40 min at 50 V onto a nitrocellulose membrane 0.45 μm (BioRad) by wet technique. A primary monoclonal mouse antibody against total MMP-2 ab86607 (Abcam) and secondary goat-anti-mouse conjugated with HRP (BioRad) were both used at a dilution of 1:1000. The blot was developed and scanned as described above. The 72 kDa MMP-2 was detected by comparison with Precision Plus Protein Standards (BioRad) and relative content was calculated on the basis of 75 kDa band intensity and expressed in arbitrary units.

### 4.7. Expression of MMP-2 Gene in Heart Tissue

Ribonucleic acid (RNA) was extracted from powdered heart tissue by phenol/chloroform technique using PureZol RNA isolation reagent (BioRad), according to the manufacturer’s instruction. Briefly, 50 μg of tissue powder was mixed with 1 mL of PureZol, and immediately homogenized by Pellet Pestle^®^ Motor (Kimble Kontes). The next steps included extraction with chloroform (Stanlab), precipitation of RNA with isopropanol (Chempur), and washing with 75% ethanol (Chempur). Purified RNA was dissolved in 50 μL of DEPC-treated water (Ambion) and its quality and concentration were assessed by NanoDrop Lite Spectrophotometer (Thermo Fisher Scientific). 500 ng of RNA was taken for reverse transcription performed using iScript™ cDNA Synthesis Kit (BioRad). Reverse transcription and subsequent real-time PCR were both performed on CFX96 Touch Real-Time PCR Detection System (BioRad). 100 ng of each cDNA template was used for both genes’ real-time amplification in duplicates, using iTaq Universal SYBR^®^ Green Supermix (BioRad) following the manufacturer’s protocol. Glyceraldehyde 3-phosphate dehydrogenase (GAPDH) served as a housekeeping gene for the normalization of MMP-2 gene expression. Primers sequences were designed as follows: GAPDH F: 5′ AGTGCCAGCCTCGTCTCATA 3′, GAPDH R: 5′ GATGGTGATGGGTTTCCCGT 3′; MMP-2 F: 5′ AGCAAGTAGACGCTGCCTTT 3′, MMP-2 R: 5′ CAGCACCTTTCTTTGGGCAC 3′. Relative fold MMP-2 gene expression was calculated according to delta-delta Ct formula.

### 4.8. Statistical Analysis

All data are expressed as mean +/− SEM. Comparisons between groups were assessed for significance by two-way analysis of variance (ANOVA) or T-test after assessment of normality of distribution. Logarithmic transformation was performed of data that had non-normal distribution to meet requirements of ANOVA analysis. The conclusions drawn after data transformation remain valid for the original data. If significance was established, posthoc analysis was done using Tukey’s test. A value of *p* < 0.05 was considered statistically significant.

## 5. Conclusions

In conclusion, our results provide the first evidence that CAR reduces mechanical dysfunction of the heart muscle, along with attenuation of MMP-2 activity and degradation of troponin I in hearts subjected to acute IR injury. These data might help to differentiate carvedilol from other β-blockers and explain a peculiarity of this drug, in terms of its anti-MMP-2 properties in the settings of acute myocardial ischemia, resulting in its better cardioprotective profile compared to other β-blockers.

## Figures and Tables

**Figure 1 pharmaceuticals-14-01276-f001:**
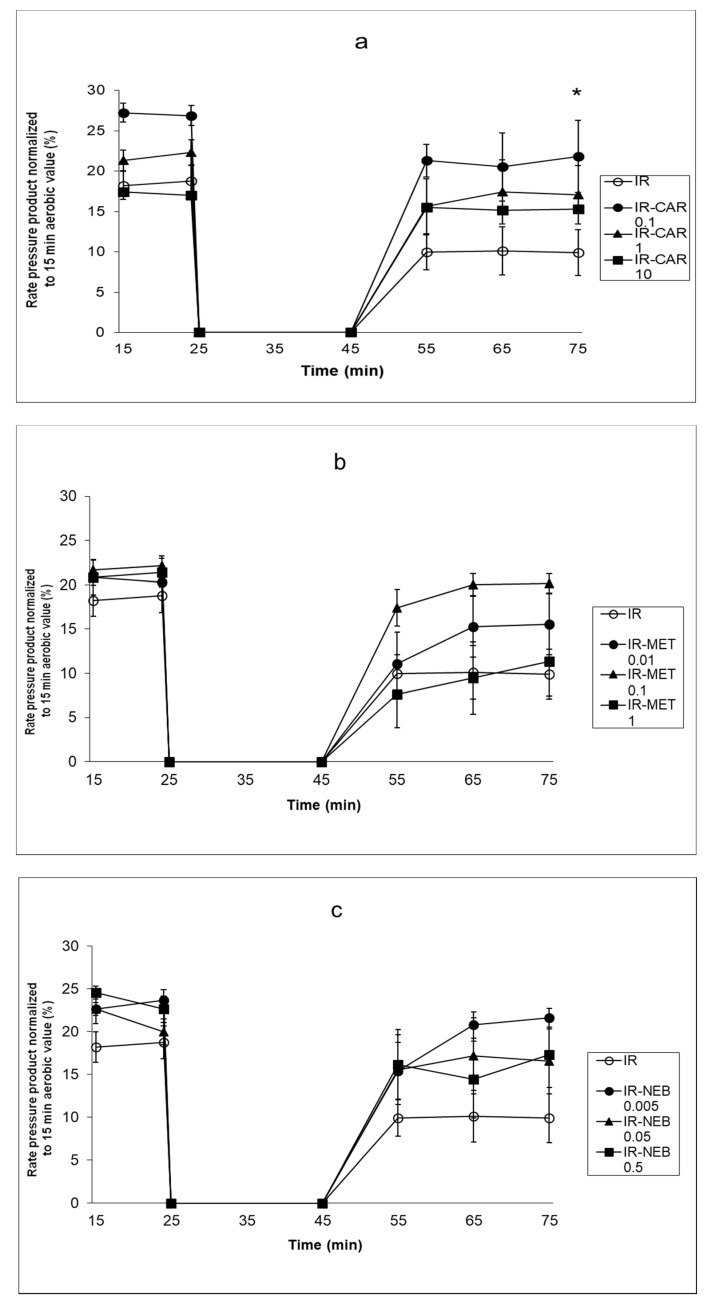
(**a**–**c**). Effect of carvedilol, metoprolol and nebivolol on cardiac mechanical function after 20 min of ischemia (**a**) Effect of carvedilol on cardiac mechanical function after 20 min of ischemia. (**b**) Effect of metoprolol on cardiac mechanical function after 20 min of ischemia. (**c**) Effect of carvedilol on cardiac mechanical function after 20 min of ischemia. The result presented as the rate-pressure product (heart rate × left ventricular developed pressure) normalized to 15 min aerobic value. C—control group, aerobically perfused hearts, IR injury—ischemia-reperfusion injury, IR-CAR 0.1– hearts from IR injury model treated with 0.1 µM carvedilol, IR-CAR 1– hearts from IR injury model treated with 1 µM carvedilol, IR-CAR 10– hearts from IR injury model treated with 10 µM carvedilol, IR-MET 0.01—hearts from IR injury model treated with 0.01 µM metoprolol, IR-MET 0.1—hearts from IR injury model treated with 0.1 µM metoprolol, IR-MET 1—hearts from IR injury model treated with 1 µM metoprolol, IR-NEB 0.005—hearts from IR injury model treated with 0.005 µM nebivolol, IR-NEB 0.05—hearts from IR injury model treated with 0.05 µM nebivolol, IR-NEB 0.5—hearts from IR injury model treated with 0.5 µM nebivolol. * IR vs IR-CAR 0.1, *p* < 0.05, *n* = 6, ANOVA.

**Figure 2 pharmaceuticals-14-01276-f002:**
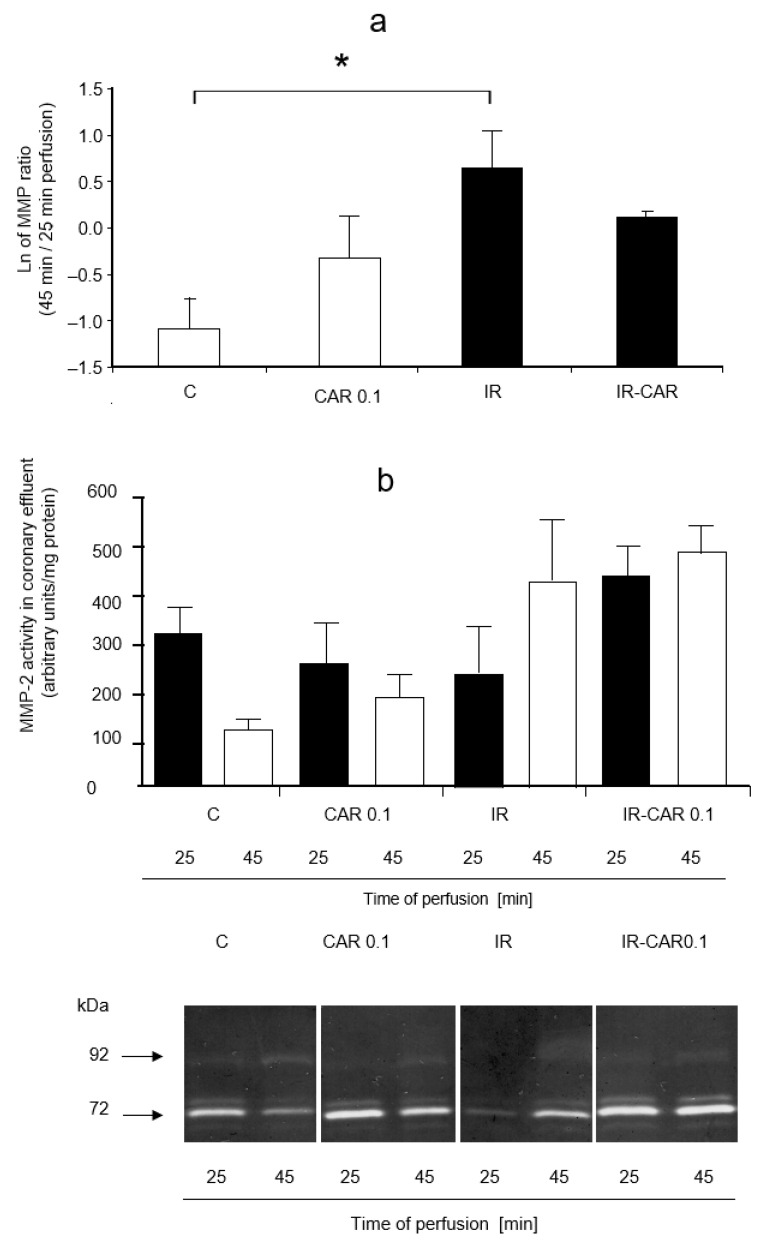
Influence of carvedilol, metoprolol and nebivolol on MMP-2 activity in coronary effluent. (**a**) Densitometric analysis of gelatinolytic MMP-2 activities in coronary effluent in samples from perfused hearts collected at different time points. (**b**) Representative zymogram showing gelatinolytic activities in coronary effluent samples from perfused hearts collected at different time points. The 72 kDa MMP-2 specific activity in coronary effluent samples was assessed by densitometric analysis (**c**) Log ratio of 45 vs. 25 min perfusion MMP-2 activities in coronary effluent in carvedilol treated groups. (**d**) Log ratio of 45 vs. 25 min perfusion MMP-2 activities in coronary effluent in metoprolol treated groups. (**e**) Log ratio of 45 vs. 25 min perfusion MMP-2 activities in coronary effluent in nebivolol treated groups. Only in the IR CAR 0.01 group there was no significant increase in MMP-2 activity. C—control group, aerobically perfused hearts, IR injury—ischemia-reperfusion injury, IR-CAR 0.1—hearts from IR injury model treated with 0.1 µM carvedilol, IR-CAR 1—hearts from IR injury model treated with 1 µM carvedilol, IR-CAR 10—hearts from IR injury model treated with 10 µM carvedilol, IR-MET 0.01—hearts from IR injury model treated with 0.01 µM metoprolol, IR-MET 0.1—hearts from IR injury model treated with 0.1 µM metoprolol, IR-MET 1—hearts from IR injury model treated with 1 µM metoprolol, IR-NEB 0.005—hearts from IR injury model treated with 0.005 µM nebivolol, IR-NEB 0.05—hearts from IR injury model treated with 0.05 µM nebivolol, IR-NEB 0.5—hearts from IR injury model treated with 0.5 µM nebivolol. * C vs IR, IR-CAR 1, IR-CAR 10, IR-MET 0.01, IR-MET 0.1, IR-MET 1, IR-NEB 0.005, IR-NEB 0.05, IR-NEB 0.5, *p* < 0.05, *n* = 6, ANOVA.

**Figure 3 pharmaceuticals-14-01276-f003:**
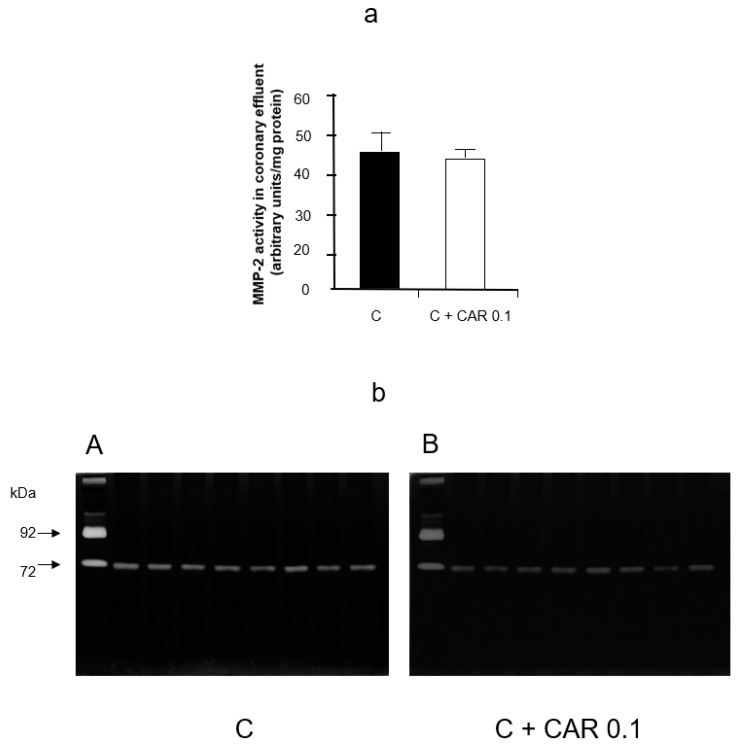
Effect of CAR on MMP-2 activity in vitro. (**a**) Densitometric analysis of gelatinolytic MMP-2 activities in control gel and gel incubated with addition of 0.1 μM CAR. (**b**) Representative zymograms of MMP-2 activity in control gel (**A**) and gel incubated with addition of 0.1 μM CAR (**B**). **C**—control group, C+ CAR 0.1—in control gel and gel incubated with addition of 0.1 μM CAR. *p* > 0.05, *n* = 8, T-test.

**Figure 4 pharmaceuticals-14-01276-f004:**
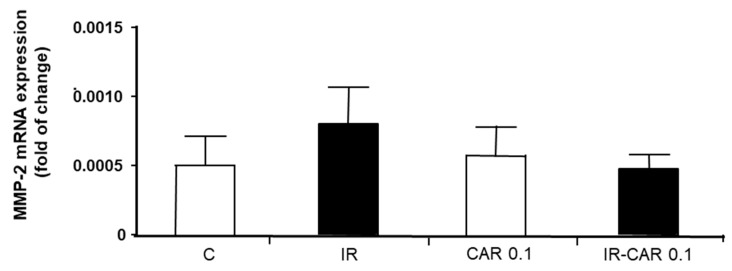
Effect of carvedilol on MMP-2 mRNA expression in hearts subjected to ischemia-reperfusion. Fold of change was calculated by delta-delta Ct formula. Glyceraldehyde 3-phosphate dehydrogenase served as a normalizing gene. C—control group, aerobically perfused hearts, IR injury—ischemia-reperfusion injury, CAR 0.1—aerobically perfused hearts treated with 0.1 µM carvedilol, IR-CAR 0.1—hearts from IR injury model treated with 0.1 µM carvedilol. *p* > 0.05, *n* = 5, ANOVA.

**Figure 5 pharmaceuticals-14-01276-f005:**
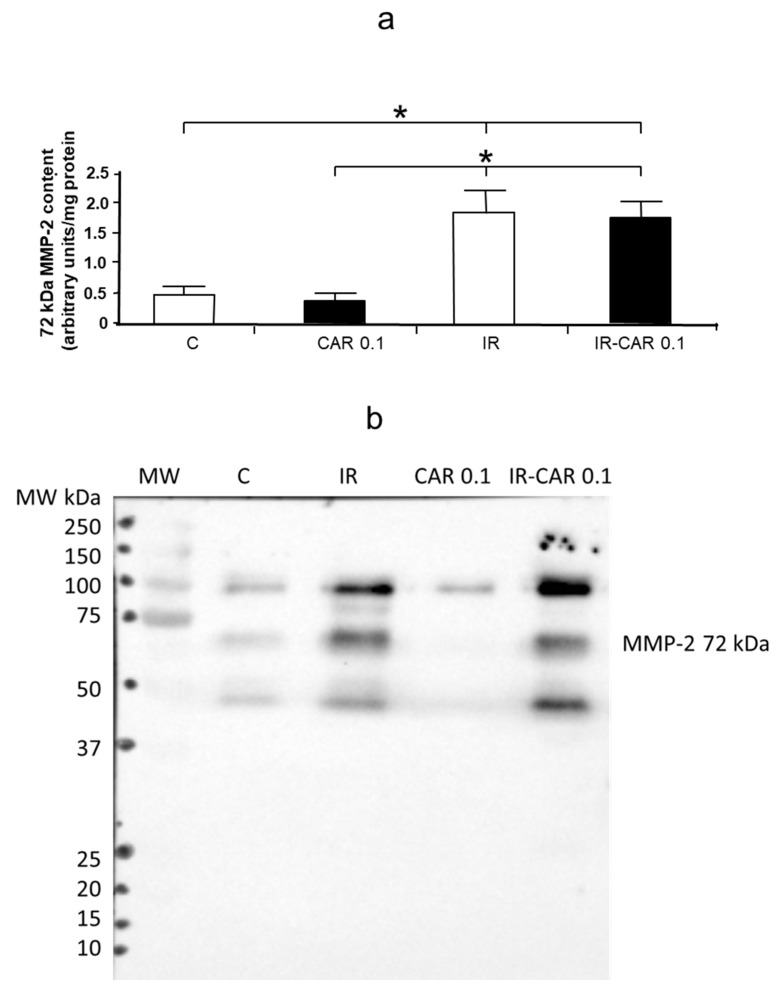
MMP-2 content in coronary effluents and heart tissue. (**a**) Densitometric analysis of MMP-2 content in coronary effluent collected at 2 min of reperfusion after 20 min ischemia in IR CAR 0.1 group as determined by Western blot. (**b**) Representative western blot of MMP-2 content in coronary effluents collected at 2 min of reperfusion after 20 min ischemia in IR CAR 0.1 group. C—control group, aerobically perfused hearts, IR injury—ischemia-reperfusion injury, CAR 0.1, aerobically perfused hearts treated with 0.1 µM carvedilol, IR-CAR 0.1—hearts from IR injury model treated with 0.1 µM carvedilol. * C vs IR, C vs IR-CAR 0.1, CAR 0.1 vs IR, CAR 0.1 vs IR-CAR 0.1, *p* < 0.05, *n* = 5, ANOVA.

**Figure 6 pharmaceuticals-14-01276-f006:**
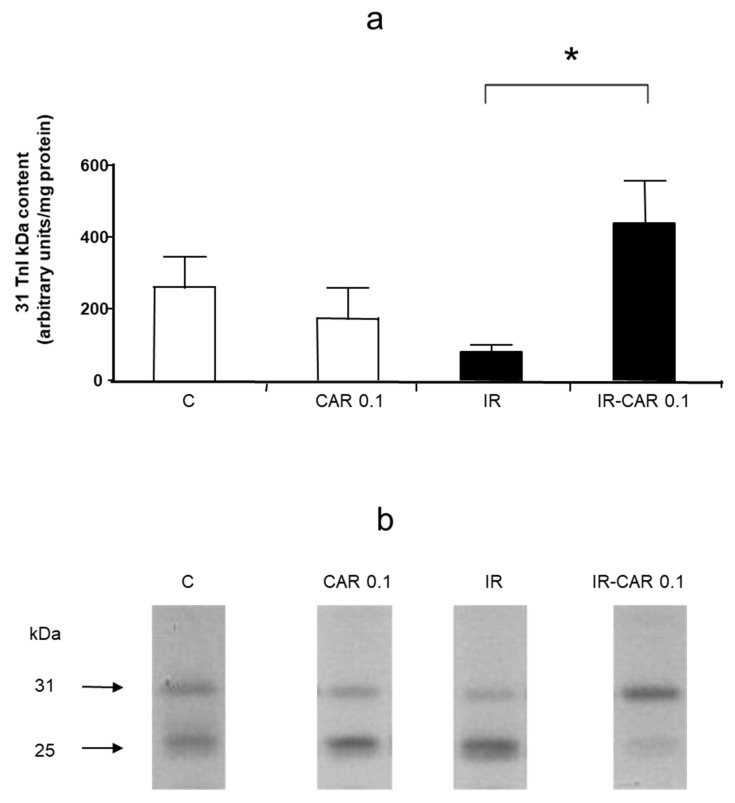
(**a**–**d**) Effect of carvedilol, metoprolol and nebivolol on troponin I content in heart tissue. (**a**) Densitometric analysis of TnI content in heart homogenates prepared at 30 min of reperfusion after 20 min ischemia in IR CAR 0.1 group as determined by Western blot. (**b**) Representative troponin I (TnI) protein level in heart homogenates prepared at the end of perfusion in IR CAR 0.1 group as determined by Western blot. (**c**) Densitometric analysis of TnI content in heart homogenates prepared at 30 min of reperfusion after 20 min ischemia in metoprolol treated hearts as determined by Western blot. (**d**) Densitometric analysis of TnI content in heart homogenates prepared at 30 min of reperfusion after 20 min ischemia in nebivolol treated hearts as determined by Western blot. Decrease of the level of 31 kDa troponin I was abolished only in the IR-CAR 0.1 μM group. C—control group, aerobically perfused hearts, IR injury—ischemia-reperfusion injury, IR-CAR 0.1—hearts from IR injury model treated with 0.1 µM carvedilol, IR-CAR 1—hearts from IR injury model treated with 1 µM carvedilol, IR-CAR 10– hearts from IR injury model treated with 10 µM carvedilol, IR-MET 0.01—hearts from IR injury model treated with 0.01 µM metoprolol, IR-MET 0.1—hearts from IR injury model treated with 0.1 µM metoprolol, IR-MET 1—hearts from IR injury model treated with 1 µM metoprolol, IR-NEB 0.005—hearts from IR injury model treated with 0.005 µM nebivolol, IR-NEB 0.05—hearts from IR injury model treated with 0.05 µM nebivolol, IR-NEB 0.5—hearts from IR injury model treated with 0.5 µM nebivolol. * IR vs IR-CAR 0.01, *p* < 0.05, *n* = 6, ANOVA.

**Figure 7 pharmaceuticals-14-01276-f007:**
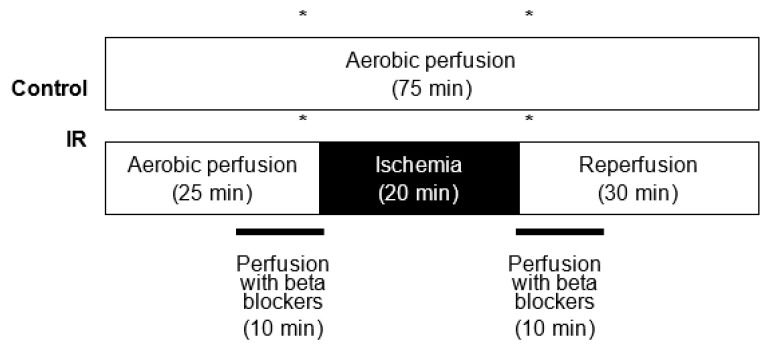
Experimental protocol for isolated heart perfusions. Detailed description in Material and methods part, “heart perfusion protocol” section. * show times when coronary effluent samples were collected during aerobic perfusion (23–25 min) and in the first 2 min of reperfusion (45–47 min).

## Data Availability

The data used to support the findings of this study are included within the article and are available from the corresponding author.

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
