# Peer review of "Matrix Metalloproteinase-2 Inhibition in Acute Ischemia-Reperfusion Heart Injury—Cardioprotective Properties of Carvedilol"

_pharmaceuticals, 2021, doi:10.3390/ph14121276_

Round 1
Reviewer 1 Report
The authors describe overlaying effects of carvedilol and ischemia-reperfusion on MMP-2 activity in coronary effluent. These observations are quite preliminary and purely descriptive, yet of potential interest as they may help to understand and further exploit a particular therapeutic potential of carvedilol in myocardial ischemia-reperfusion.
However, in its present form, the paper is unfit for publication for three reasons:
- It remains unclear, whether the described alterations of gross MMP-2 activity in corornary effluent are due to changes in enzyme expression (i.e. induction vs. repression), enzyme localisation (i.e release vs. retention) or enzyme activity (i.e. activation vs. inhibition). This issue needs to be addressed by additional experiments.
- The authors claim that the observed effects are specific for carvedilol and not seen with any other beta-blocker currently in clinical use. However, they show no data obtained with any beta-blocker other than carvedilol. So how is the above claim supported experimentally?
- I had problems understanding what was done, how it was done, why it was done. Data presentation in figures must clarified and simplified as follows: (i) Each figure must have a legend. Legends need to be sufficiently clear and detailed to render the figures self-explaining (without the results text). (ii) Authors should refrain from complicated data-transformations such as in Fig. 2A, which are not really needed for understanding the experiment.
Concerns with missing experiments and controls:
It remains unclear, what kind of effects of IR ± CAR on myocardial MMP-2 one is looking at: Induction/repression, release/retention, activation/inhibition of MMP-2? To clarify this crucial question the following data needs to be furnished in addition:
- Effect of CAR on MMP-2 activity in vitro or ex vivo.
- Effect of IR ± CAR on MMP-message (mRNA) in myocardial extracts.
- Effect of IR ± CAR on MMP-mass (protein) in coronary effluents and myocardial extracts
Concerns with presentation:
I do not understand sentence in line 101: IR injury did not cause significant change in 72 kDa MMP-2 activity in heart tissue, neither in control nor in study groups (data not shown). I was under the impression that the whole point of the paper was to demonstrate the opposite.
I do not understand Fig 2 and results text:
Fig 2A: What is the purpose of this plot? What do the authors wish to demonstrate?
Fig 2B/C: Raw data and quantitative readouts of same data as in 1A? If so, the following two questions arise: (i) CAR per se (in the absence of IR) seems to have an effect on the 45 min-perfusion value. Why is that and why does the text line 97 state the opposite? (ii) IR seems to decrease 25 min value of MMP-release. Why is that and why is it not addressed in the text?
Fig. 3 has no legend. Is one looking at Trop I in heart tissue or in coronary effluent? Why is the level following IR+CAR higher than in the control?
Author Response
Reviewer #1
The authors describe overlaying effects of carvedilol and ischemia-reperfusion on MMP-2 activity in coronary effluent. These observations are quite preliminary and purely descriptive, yet of potential interest as they may help to understand and further exploit a particular therapeutic potential of carvedilol in myocardial ischemia-reperfusion. However, in its present form, the paper is unfit for publication for three reasons:
Response:
We thank the Reviewer #1 for careful review and constructive comments which helped to improve the manuscript. Please find below our answers to the specific comments.
It remains unclear, whether the described alterations of gross MMP-2 activity in coronary effluent are due to changes in enzyme expression (i.e. induction vs. repression), enzyme localisation (i.e release vs. retention) or enzyme activity (i.e. activation vs. inhibition). This issue needs to be addressed by additional experiments.
Response:
We thank Reviewer #1 for bringing up this important matter. Indeed, based on our results we certainly cannot say if the described alterations of gross MMP-2 activity in coronary effluent are due to changes in enzyme expression, enzyme localisation or enzyme activity. We agree that addressing this issue by additional experiments suggested by the Reviewer and adding this to the paper would certainly add weight to our manuscript. Unfortunately, time given to us for revision of the manuscript is too short for performing new experiments and incorporating them to the manuscript. The Reviewer’s comment was therefore acknowledged and included as a study limitation in the revised manuscript (Microsoft Word version: page 11, lines 530-540, PDF version: page 11, lines 370-380):
“The main limitation of our study is that, based on our results, we cannot directly say if the described alterations of MMP-2 activity in coronary effluent are due to changes in enzyme expression, enzyme localization or enzyme activity. Due to the short duration of ischemia, we can exclude, with high probability, changes at the transcriptional level. The increase in coronary effluent MMP-2 activity due to IR was accompanied by troponin I degradation in heart tissue, while a decrease in MMP-2 activity in the effluent from hearts treated with carvedilol was accompanied by a decrease in tissue troponin I degradation. These results indirectly suggest that the changes of MMP-2 activity resulted more from its activation in heart tissue and subsequent release into the coronary effluent in the settings of IR than from changes in enzyme localization. Further research is clearly needed to address our findings”.
The authors claim that the observed effects are specific for carvedilol and not seen with any other beta-blocker currently in clinical use. However, they show no data obtained with any beta-blocker other than carvedilol. So how is the above claim supported experimentally?
Response:
As per Reviewer #1 suggestion the data obtained with metoprolol and carvedilol are now added to the Results section which was rewritten according to other suggestions (Microsoft Word version: page 2, lines 90-95, page 4, lines 135-147, page 7, lines 252-255, PDF version: page 2, lines 88-93, page 4, lines 135-147, page 7, lines 226-229 ). Incorporating new data required adding new figures: Fig. 1b, Fig.1c. (Microsoft Word version: page 3, lines 117-118, PDF version: page 3, lines 117-118). Fig 1 after adding missing data is now Fig. 1a (Microsoft Word version: page 3, lines 116, PDF version: page 3, lines 116). Fig. 2 was changed according suggestions below and of Reviewer #2 (both Microsoft Word and PDF versions: pages 5 and 6, lines 148-150). New data were added to Fig 3 which is now divided into Fig. 3a, Fig. 3b and Fig. 3c and Fig. 3 d (Microsoft Word version: pages 8 and 9, lines 256-264, PDF version: pages 8 and 9, lines 230-238). New Figure legends were incorporated to the manuscript (Microsoft Word version: page 4, lines 120-133, page 6, lines 232-250, pages 9 and 10, lines 266-285, PDF version: page 4, lines 120-133, page 6, lines 206-224, pages 9 and 10, lines 240-259 )
“IR injury caused a significant reduction in the heart rate and mechanical function recovery expressed as LVDP and RPP in the reperfusion period (Fig. 1 a-c). CAR 0.1 μM significantly improved the recovery of mechanical function at the end of reperfusion period in comparison to the control IR injury group (P < 0.01; Fig. 1a). The higher CAR concentrations, as well NEB and MET in all tested concentrations, did not significantly improve the recovery after IR injury (Fig. 1 a-c).”
“The differences when comparing MMP-2 activities in the coronary effluent between carvedilol treated groups were not significant (Figure 2a). Also at higher CAR concentrations, as well as with NEB and MET at all tested concentrations, there were no significant differences between groups (data not shown).
Nevertheless, a significant increase of MMP-2 activity in the coronary effluent, calculated as the ratio of 45 and 25 min perfusion time values was seen in IR injury group as well as CAR 1 and CAE 10 groups in comparison with C group (P < 0.0001; Fig. 2 c-e). This increase in MMP-2 activity was significantly lower in the IR injury with CAR 0.1 μM group (Fig. 2c). NEB and MET in all tested concentrations had no significant influence on the increase of MMP-2 activity evoked by IR which was significantly higher than in the C group (P < 0.0001, Fig, 2 d-e).”
" Analysis of troponin I levels in hearts at the end of perfusion showed that IR injury caused a 73% decrease of the level of 31 kDa troponin I which was abolished in the IR-CAR 0.1 μM group (P < 0.05; Fig. 3 a-b). This was not observed NEB and MET in all tested concentrations (Fig 3 c-d).”
“Fig.1 a-c. Effect of carvedilol, metoprolol and nebivolol on cardiac mechanical function after 20 minutes of ischemia a. Effect of carvedilol on cardiac mechanical function after 20 minutes of ischemia. b. Effect of metoprolol on cardiac mechanical function after 20 minutes of ischemia. c. Effect of carvedilol on cardiac mechanical function after 20 minutes of ischemia. Result presented as the rate-pressure product (heart rate × left ventricular developed pressure) normalized to 15 minutes aerobic value. C – control group, aerobically perfused hearts, IR injury – ischemia-reperfusion injury, IR-CAR 0.1– hearts from IR injury model treated with 0.1 µM carvedilol, IR-CAR 1– hearts from IR injury model treated with 1 µM carvedilol, IR-CAR 10– hearts from IR injury model treated with 10 µM carvedilol, IR-MET 0.01 – hearts from IR injury model treated with 0.01 µM metoprolol, IR-MET 0.1 – hearts from IR injury model treated with 0.1 µM metoprolol, IR-MET 1 – hearts from IR injury model treated with 1 µM metoprolol, IR-NEB 0.005 – hearts from IR injury model treated with 0.005 µM nebivolol, IR-NEB 0.05 – hearts from IR injury model treated with 0.05 µM nebivolol, IR-NEB 0.5 – hearts from IR injury model treated with 0.5 µM nebivolol * IR vs IR-CAR 0.1, p<0.05, n=6, ANOVA.”
“Figure 2. Influence of carvedilol, metoprolol and nebivolol on MMP-2 activity in coronary effluent. a. Densitometric analysis of gelatinolytic MMP-2 activities in coronary effluent in samples from perfused hearts collected at different time points. b. Representative zymogram showing gelatinolytic activities in coronary effluent samples from perfused hearts collected at different time points. The 72 kDa MMP-2 specific activity in coronary effluent samples was assessed by densitometric analysis c. Log ratio of 45 vs. 25 min perfusion MMP-2 activities in coronary effluent in carvedilol treated groups. d. Log ratio of 45 vs. 25 min perfusion MMP-2 activities in coronary effluent in metoprolol treated groups. e. Log ratio of 45 vs. 25 min perfusion MMP-2 activities in coronary effluent in nebivolol treated groups. Only in the IR CAR 0.01 group there was no significant increase in MMP-2 activity.
C – control group, aerobically perfused hearts, IR injury – ischemia-reperfusion injury, IR-CAR 0.1– hearts from IR injury model treated with 0.1 µM carvedilol, IR-CAR 1– hearts from IR injury model treated with 1 µM carvedilol, IR-CAR 10– hearts from IR injury model treated with 10 µM carvedilol, IR-MET 0.01 – hearts from IR injury model treated with 0.01 µM metoprolol, IR-MET 0.1 – hearts from IR injury model treated with 0.1 µM metoprolol, IR-MET 1 – hearts from IR injury model treated with 1 µM metoprolol, IR-NEB 0.005 – hearts from IR injury model treated with 0.005 µM nebivolol, IR-NEB 0.05 – hearts from IR injury model treated with 0.05 µM nebivolol, IR-NEB 0.5 – hearts from IR injury model treated with 0.5 µM nebivolol. *C vs IR, IR-CAR 1, IR-CAR 10, IR-MET 0.01, IR-MET 0.1, IR-MET 1, IR-NEB 0.005, IR-NEB 0.05, IR-NEB 0.5, p<0.05, n=6, ANOVA.”
“Fig.3 a-d. Effect of carvedilol, metoprolol and nebivolol on troponin I content in heart tissue. a. Densitometric analysis of TnI content in heart homogenates prepared at 30 min of reperfusion after 20 min ischemia in IR CAR 0.1 group as determined by Western blot. b. Representative troponin I (TnI) protein level in heart homogenates prepared at the end of perfusion in IR CAR 0.1group as determined by Western blot. c. Densitometric analysis of TnI content in heart homogenates prepared at 30 min of reperfusion after 20 min ischemia in metoprolol treated hearts as de-termined by Western blot. Densitometric analysis of TnI content in heart homogenates prepared at 30 min of reperfusion after 20 min ischemia in metoprolol treated hearts in nebivolol treated hearts as de-termined by Western blot. Decrease of the level of 31 kDa troponin I was abolished only in the IRI-CAR 0.1 μM group.
C – control group, aerobically perfused hearts, IR injury – ischemia-reperfusion injury, IR-CAR 0.1– hearts from IR injury model treated with 0.1 µM carvedilol, IR-CAR 1– hearts from IR injury model treated with 1 µM carvedilol, IR-CAR 10– hearts from IR injury model treated with 10 µM carvedilol, IR-MET 0.01 – hearts from IR injury model treated with 0.01 µM metoprolol, IR-MET 0.1 – hearts from IR injury model treated with 0.1 µM metoprolol, IR-MET 1 – hearts from IR injury model treated with 1 µM metoprolol, IR-NEB 0.005 – hearts from IR injury model treated with 0.005 µM nebivolol, IR-NEB 0.05 – hearts from IR injury model treated with 0.05 µM nebivolol, IR-NEB 0.5 – hearts from IR injury model treated with 0.5 µM nebivolol * IR vs IR-CAR 0.01, p<0.05, n=6, ANOVA.
I had problems understanding what was done, how it was done, why it was done. Data presentation in figures must clarified and simplified as follows: (i) Each figure must have a legend. Legends need to be sufficiently clear and detailed to render the figures self-explaining (without the results text). (ii) Authors should refrain from complicated data-transformations such as in Fig. 2A, which are not really needed for understanding the experiment.
Response: We thank the Reviewer #1 for pointing this out. We have simplified and clarified legends to be more precise
(Microsoft Word version: page 4, lines 120-133, page 6, lines 232-250, pages 9 and 10, lines 266-285, PDF version: page 4, lines 120-133, page 6, lines 206-224, pages 9 and 10, lines 240-259 )
. Logarithmic transformation of data was needed due to its non-normal distribution to meet requirements of ANOVA analysis. The conclusions drawn after data transformation remain valid for the original data. To clarify this issue, the Materials and Methods section was changed as follows (Microsoft Word version: page 13, lines 655-657, PDF version: page 13, lines 495-497 ). The ratio of 45 vs. 25 min perfusion MMP-2 activities in coronary effluent reflects the ability of the tested drugs to reduce MMP-2 activity. This parameter was used because the differences when comparing MMP-2 activities between groups were not significant. This was also clarified in the Results section (Both Microsoft Word and PDF versions: page 4 , lines 135-145)
“Logarithmic transformation of data was performed due to its non-normal distribution to meet requirements of ANOVA analysis. The conclusions drawn after data transformation remain valid for the original data.”
“The differences when comparing MMP-2 activities in the coronary effluent between carvedilol treated groups were not significant (Figure 2a). Also at higher CAR concentrations, as well as with NEB and MET at all tested concentrations, there were no significant differences between groups (data not shown).
Nevertheless, a significant increase of MMP-2 activity in the coronary effluent, calculated as the ratio of 45 and 25 min perfusion time values was seen in IR injury group as well as CAR 1 and CAE 10 groups in comparison with C group (P < 0.0001; Fig. 2 c-e). This increase in MMP-2 activity was significantly lower in the IR injury with CAR 0.1 μM group (Fig. 2c). NEB and MET in all tested concentrations had no significant influence on the increase of MMP-2 activity evoked by IR which was significantly higher than in the C group (P < 0.0001, Fig, 2 d-e).”
Concerns with missing experiments and controls:
It remains unclear, what kind of effects of IR ± CAR on myocardial MMP-2 one is looking at: Induction/repression, release/retention, activation/inhibition of MMP-2? To clarify this crucial question the following data needs to be furnished in addition:
Effect of CAR on MMP-2 activity in vitro or ex vivo.
Effect of IR ± CAR on MMP-message (mRNA) in myocardial extracts.
Effect of IR ± CAR on MMP-mass (protein) in coronary effluents and myocardial extracts
Response: We thank Reviewer #1 for the suggestions. As we previously stated, it is a very important question and the main limitation of the study. Unfortunately, time given to us for revision of the manuscript is too short for performing new experiments and incorporating them to the manuscript. We have now acknowledged this as a study limitation and suggested it as a topic for further research in the Discussion section of the revised manuscript (Microsoft Word version: page 11, lines 530-540, PDF version: page 11, lines 370-380).
“The main limitation of our study is that, based on our results, we cannot directly say if the described alterations of MMP-2 activity in coronary effluent are due to changes in enzyme expression, enzyme localization or enzyme activity. Due to the short duration of ischemia, we can exclude, with high probability, changes at the transcriptional level. The increase in coronary effluent MMP-2 activity due to IR was accompanied by troponin I degradation in heart tissue, while a decrease in MMP-2 activity in the effluent from hearts treated with carvedilol was accompanied by a decrease in tissue troponin I degradation. These results indirectly suggest that the changes of MMP-2 activity resulted more from its activation in heart tissue and subsequent release into the coronary effluent in the settings of IR than from changes in enzyme localization. Further re-search is clearly needed to address our findings.”.
Concerns with presentation:
I do not understand sentence in line 101: IR injury did not cause significant change in 72 kDa MMP-2 activity in heart tissue, neither in control nor in study groups (data not shown). I was under the impression that the whole point of the paper was to demonstrate the opposite.
Response: We thank Reviewer #1 for pointing out that our statement could be understood in this way. We agree with the Reviewer that lack of significant changes in MMP-2 tissue activity do not support our hypothesis. However, the tissue MMP-2 activity was assessed only at 30 minutes of reperfusion, but not within the first two minutes. As a result of reperfusion injury following myocardial ischemia we showed a rapid and enhanced release of MMP-2 into the coronary effluent which peaked within the first 2 min of reperfusionas shown by [Wang et al., Circulation. 2002;106(12): 1543-1549] . As increased release of activated MMP-2 results from its intracellular activation, consequently, MMP-2 tissue activity at the beginning of reperfusion should be increased. The lack of increased MMP-2 activity in heart homogenates at the end of reperfusion is due to its release from isolated hearts as shown by Cheung et al. [Circulation. 2000; 101(15): 1833-1839]. Again, troponin I levels in heart tissue may serve as indirect evidence of MMP activation [Wang et al., Circulation. 2002;106(12): 1543-1549]. In spite of this limitation we suggest that the effects of carvedilol on postischemic cardiac contractility and troponin levels we observe may be due to its ability to inhibit MMP-2 activation and subsequent release. We have already discussed this issue as a main limitation of the study but we have now also changed the result section as follows (Both Microsoft Word and PDF versions: page 4, lines 146-147):
“MMP-2 activity in the heart tissue was assessed only at the end of the experiment showing no significant changes between groups (data not shown)
I do not understand Fig 2 and results text:
Fig 2A: What is the purpose of this plot? What do the authors wish to demonstrate?
Response: The ratio of 45 vs. 25 min perfusion MMP-2 activities in coronary effluent reflects the ability of tested drugs to reduce MMP-2 activity. This parameter was used because the differences when comparing MMP-2 activities between groups were not significant. This was also clarified in Result section (Both Microsoft Word and PDF versions: page 4 , lines 135-145)
“The differences when comparing MMP-2 activities in the coronary effluent between carvedilol treated groups were not significant (Figure 2a). Also at higher CAR concentrations, as well as with NEB and MET at all tested concentrations, there were no significant differences between groups (data not shown).
Nevertheless, a significant increase of MMP-2 activity in the coronary effluent, calculated as the ratio of 45 and 25 min perfusion time values was seen in IR injury group as well as CAR 1 and CAE 10 groups in comparison with C group (P < 0.0001; Fig. 2 c-e). This increase in MMP-2 activity was significantly lower in the IR injury with CAR 0.1 μM group (Fig. 2c). NEB and MET in all tested concentrations had no significant influence on the increase of MMP-2 activity evoked by IR which was significantly higher than in the C group (P < 0.0001, Fig, 2 d-e).”
Fig 2B/C: Raw data and quantitative readouts of same data as in 1A? If so, the following two questions arise: (i) CAR per se (in the absence of IR) seems to have an effect on the 45 min-perfusion value. Why is that and why does the text line 97 state the opposite? (ii) IR seems to decrease 25 min value of MMP-release. Why is that and why is it not addressed in the text?
Response: We thank the Reviewer #1 for these observations. The differences between values were not significant. Now this is clarified in the Results section (Both Microsoft Word and PDF versions: page 4, lines 135-136)
“The differences when comparing MMP-2 activities in the coronary effluent between carvedilol treated groups were not significant (Figure 2b).”
Fig. 3 has no legend. Is one looking at Trop I in heart tissue or in coronary effluent? Why is the level following IR+CAR higher than in the control?
Response: We thank the Reviewer for pointing it out. We would like to apologize for our carelessness. The legend was incorporated in the text in Word format, but unfortunately it was lost during reformatting into the PDF. We were not careful enough with this process. Now the legend is incorporated in both Word and PDF versions (Microsoft Word version: pages 9 and 10 , lines 266-285. PDF version: pages 9 and 10, lines 240-259).

Reviewer 2 Report
Comments to authors:
The authors studied the effects of beta adrenergic receptor blockers on different MMPs. It was hypothesized that carvedilol, nebivolol, and/or metoprolol could reduce MMPs activities in myocytes, providing a protection against myocardial dysfunction related to IR-induced injury. Isolated rat hearts were used for studies, and it was found that carvedilol improved cardiac recovery during the reperfusion period in connection with the activation and release of MMP-2 in the coronary effluent. It was concluded that the cardioprotective effect of carvedilol may be mediated by MMP-2 activation.
Comments to the authors:
Is it possible that there is a significant difference between the C and I/R groups, and no significant difference can be seen between the C and IR-CAR 0.1 microM groups (Fig. 2A)?
What is the reason that no degrees of statistical significance were detected in MMP-2 activities in Fig. 2B?
The level of statistical significance should be also indicated and written for the Legend in Figure 2 (*, and p value).
How about Fig.3. in the manuscript!? Has this Figure got a number and legend? The number and legend for Figure 3. is completely missing. It has to be completed.
How about the results of nebivolol and metoprolol!? No any results have been found by this reviewer in the manuscript, although the authors described in the Abstract that “The effects of carvedilol, nebivolol, or metoprolol were evaluated in hearts subjected to IR injury.” Several doses of these drugs (dose response) are mentioned and no results presented at all.
Several concentrations of carvedilol also described in the text, and a single concentration (0.1 micromole) of this drug was only studied.
This reviewer thinks that this manuscript needs a substantial and very extensive revision before the acceptance for publication.
The scientific English and the presented (or not presented but described) results must be substantially improved and very carefully checked by the Canadian coauthor (CA, Alberta), as a native English speaker, as well.
Author Response
Reviewer # 2
The authors studied the effects of beta adrenergic receptor blockers on different MMPs. It was hypothesized that carvedilol, nebivolol, and/or metoprolol could reduce MMPs activities in myocytes, providing a protection against myocardial dysfunction related to IR-induced injury. Isolated rat hearts were used for studies, and it was found that carvedilol improved cardiac recovery during the reperfusion period in connection with the activation and release of MMP-2 in the coronary effluent. It was concluded that the cardioprotective effect of carvedilol may be mediated by MMP-2 activation.
Response: We thank the Reviewer for their thoughtful and thorough review and believe their input is invaluable in improving our manuscript.
Comments to the authors:
Is it possible that there is a significant difference between the C and I/R groups, and no significant difference can be seen between the C and IR-CAR 0.1 microM groups (Fig. 2A)?
Response: We thank the Reviewer for this question. The post hock test results for T45/T25 are as follows: for C s IR p=0,000365, for C vs IR-CAR 0.1 p=0,130156.
What is the reason that no degrees of statistical significance were detected in MMP-2 activities in Fig. 2B?
Response: We thank the Reviewer for this question. The data in this figure are from the raw, non-log-transformed summary data. Therefore it was not possible to do ANOVA on these data as they were not normally distributed.
The level of statistical significance should be also indicated and written for the Legend in Figure 2 (*, and p value).
Response: We thank the Reviewer #2 for pointing this out. The missing parameters were added to the figure and figure legends (both Microsoft Word and PDF versions: pages 5 and 6, lines 148-150).
How about Fig.3. in the manuscript!? Has this Figure got a number and legend? The number and legend for Figure 3. is completely missing. It has to be completed.
Responce: We thank the Reviewer #2 for pointing it out. We would like to apologize for our carelessness. The legend was incorporated in the text in Word format, but unfortunately it was lost during reformatting into the PDF. We were not careful enough with this process. Now the legend is incorporated in both Word and PDF versions (Microsoft Word version: pages 9 and 10, lines 266-285, PDF version: pages 9 and 10, lines 240-259).
“Fig.3 a-d. Effect of carvedilol, metoprolol and nebivolol on troponin I content in heart tissue. a. Densitometric analysis of TnI content in heart homogenates prepared at 30 min of reperfusion after 20 min ischemia in IR CAR 0.1 group as determined by Western blot. b. Representative troponin I (TnI) protein level in heart homogenates prepared at the end of perfusion in IR CAR 0.1group as determined by Western blot. c. Densitometric analysis of TnI content in heart homogenates prepared at 30 min of reperfusion after 20 min ischemia in metoprolol treated hearts as de-termined by Western blot. Densitometric analysis of TnI content in heart homogenates prepared at 30 min of reperfusion after 20 min ischemia in metoprolol treated hearts in nebivolol treated hearts as de-termined by Western blot. Decrease of the level of 31 kDa troponin I was abolished only in the IRI-CAR 0.1 μM group.
C – control group, aerobically perfused hearts, IR injury – ischemia-reperfusion injury, IR-CAR 0.1– hearts from IR injury model treated with 0.1 µM carvedilol, IR-CAR 1– hearts from IR injury model treated with 1 µM carvedilol, IR-CAR 10– hearts from IR injury model treated with 10 µM carvedilol, IR-MET 0.01 – hearts from IR injury model treated with 0.01 µM metoprolol, IR-MET 0.1 – hearts from IR injury model treated with 0.1 µM metoprolol, IR-MET 1 – hearts from IR injury model treated with 1 µM metoprolol, IR-NEB 0.005 – hearts from IR injury model treated with 0.005 µM nebivolol, IR-NEB 0.05 – hearts from IR injury model treated with 0.05 µM nebivolol, IR-NEB 0.5 – hearts from IR injury model treated with 0.5 µM nebivolol * IR vs IR-CAR 0.01, p<0.05, n=6, ANOVA.
How about the results of nebivolol and metoprolol!? No any results have been found by this reviewer in the manuscript, although the authors described in the Abstract that “The effects of carvedilol, nebivolol, or metoprolol were evaluated in hearts subjected to IR injury.” Several doses of these drugs (dose response) are mentioned and no results presented at all.
Response: We thank the Reviewer #2 for this suggestion. The data obtained with metoprolol and carvedilol are now added to the Results section which was rewritten according to other suggestions (Microsoft Word version: page 2, lines 90-95, page 4, lines 135-147, page 7, lines 252-255, PDF version: page 2, lines 88-93, page 4, lines 135-147, page 7, lines 226-229 ). Incorporating new data required adding new figures: Fig. 1b, Fig.1c. (Microsoft Word version: page 3, lines 117-118, PDF version: page 3, lines 117-118). Fig 1 after adding missing data is now Fig. 1a (Microsoft Word version: page 3, lines 116, PDF version: page 3, lines 116). Fig. 2 was changed according suggestions below and of Reviewer #2 (both Microsoft Word and PDF versions: pages 5 and 6, lines 148-150). New data were added to Fig 3 which is now divided into Fig. 3a, Fig. 3b and Fig. 3c and Fig. 3 d (Microsoft Word version: pages 8 and 9, lines 256-264, PDF version: pages 8 and 9, lines 230-238). New Figure legends were incorporated to the manuscript (Microsoft Word version: page 4, lines 120-133, page 6, lines 232-250, pages 9 and 10, lines 266-285, PDF version: page 4, lines 120-133, page 6, lines 206-224, pages 9 and 10, lines 240-259 )
“IR injury caused a significant reduction in the heart rate and mechanical function recovery expressed as LVDP and RPP in the reperfusion period (Fig. 1 a-c). CAR 0.1 μM significantly improved the recovery of mechanical function at the end of reperfusion period in comparison to the control IR injury group (P < 0.01; Fig. 1a). The higher CAR concentrations, as well NEB and MET in all tested concentrations, did not significantly improve the recovery after IR injury (Fig. 1 a-c).”
“The differences when comparing MMP-2 activities in the coronary effluent between carvedilol treated groups were not significant (Figure 2a). Also at higher CAR concentrations, as well as with NEB and MET at all tested concentrations, there were no significant differences between groups (data not shown).
Nevertheless, a significant increase of MMP-2 activity in the coronary effluent, calculated as the ratio of 45 and 25 min perfusion time values was seen in IR injury group as well as CAR 1 and CAE 10 groups in comparison with C group (P < 0.0001; Fig. 2 c-e). This increase in MMP-2 activity was significantly lower in the IR injury with CAR 0.1 μM group (Fig. 2c). NEB and MET in all tested concentrations had no significant influence on the increase of MMP-2 activity evoked by IR which was significantly higher than in the C group (P < 0.0001, Fig, 2 d-e).”
" Analysis of troponin I levels in hearts at the end of perfusion showed that IR injury caused a 73% decrease of the level of 31 kDa troponin I which was abolished in the IRI-CAR 0.1 μM group (P < 0.05; Fig. 3 a-b). This was not observed NEB and MET in all tested concentrations (Fig 3 c-d).”
“Fig.1 a-c. Effect of carvedilol, metoprolol and nebivolol on cardiac mechanical function after 20 minutes of ischemia a. Effect of carvedilol on cardiac mechanical function after 20 minutes of ischemia. b. Effect of metoprolol on cardiac mechanical function after 20 minutes of ischemia. c. Effect of carvedilol on cardiac mechanical function after 20 minutes of ischemia. Result presented as the rate-pressure product (heart rate × left ventricular developed pressure) normalized to 15 minutes aerobic value. C – control group, aerobically perfused hearts, IR injury – ischemia-reperfusion injury, IR-CAR 0.1– hearts from IR injury model treated with 0.1 µM carvedilol, IR-CAR 1– hearts from IR injury model treated with 1 µM carvedilol, IR-CAR 10– hearts from IR injury model treated with 10 µM carvedilol, IR-MET 0.01 – hearts from IR injury model treated with 0.01 µM metoprolol, IR-MET 0.1 – hearts from IR injury model treated with 0.1 µM metoprolol, IR-MET 1 – hearts from IR injury model treated with 1 µM metoprolol, IR-NEB 0.005 – hearts from IR injury model treated with 0.005 µM nebivolol, IR-NEB 0.05 – hearts from IR injury model treated with 0.05 µM nebivolol, IR-NEB 0.5 – hearts from IR injury model treated with 0.5 µM nebivolol * IR vs IR-CAR 0.1, p<0.05, n=6, ANOVA.”
“Figure 2. Influence of carvedilol, metoprolol and nebivolol on MMP-2 activity in coronary effluent. a. Densitometric analysis of gelatinolytic MMP-2 activities in coronary effluent in samples from perfused hearts collected at different time points. b. Representative zymogram showing gelatinolytic activities in coronary effluent samples from perfused hearts collected at different time points. The 72 kDa MMP-2 specific activity in coronary effluent samples was assessed by densitometric analysis c. Log ratio of 45 vs. 25 min perfusion MMP-2 activities in coronary effluent in carvedilol treated groups. d. Log ratio of 45 vs. 25 min perfusion MMP-2 activities in coronary effluent in metoprolol treated groups. e. Log ratio of 45 vs. 25 min perfusion MMP-2 activities in coronary effluent in nebivolol treated groups. Only in the IR CAR 0.01 group there was no significant increase in MMP-2 activity.
C – control group, aerobically perfused hearts, IR injury – ischemia-reperfusion injury, IR-CAR 0.1– hearts from IR injury model treated with 0.1 µM carvedilol, IR-CAR 1– hearts from IR injury model treated with 1 µM carvedilol, IR-CAR 10– hearts from IR injury model treated with 10 µM carvedilol, IR-MET 0.01 – hearts from IR injury model treated with 0.01 µM metoprolol, IR-MET 0.1 – hearts from IR injury model treated with 0.1 µM metoprolol, IR-MET 1 – hearts from IR injury model treated with 1 µM metoprolol, IR-NEB 0.005 – hearts from IR injury model treated with 0.005 µM nebivolol, IR-NEB 0.05 – hearts from IR injury model treated with 0.05 µM nebivolol, IR-NEB 0.5 – hearts from IR injury model treated with 0.5 µM nebivolol. *C vs IR, IR-CAR 1, IR-CAR 10, IR-MET 0.01, IR-MET 0.1, IR-MET 1, IR-NEB 0.005, IR-NEB 0.05, IR-NEB 0.5, p<0.05, n=6, ANOVA.”
“Fig.3 a-d. Effect of carvedilol, metoprolol and nebivolol on troponin I content in heart tissue. a. Densitometric analysis of TnI content in heart homogenates prepared at 30 min of reperfusion after 20 min ischemia in IR CAR 0.1 group as determined by Western blot. b. Representative troponin I (TnI) protein level in heart homogenates prepared at the end of perfusion in IR CAR 0.1group as determined by Western blot. c. Densitometric analysis of TnI content in heart homogenates prepared at 30 min of reperfusion after 20 min ischemia in metoprolol treated hearts as de-termined by Western blot. Densitometric analysis of TnI content in heart homogenates prepared at 30 min of reperfusion after 20 min ischemia in metoprolol treated hearts in nebivolol treated hearts as de-termined by Western blot. Decrease of the level of 31 kDa troponin I was abolished only in the IRI-CAR 0.1 μM group.
C – control group, aerobically perfused hearts, IR injury – ischemia-reperfusion injury, IR-CAR 0.1– hearts from IR injury model treated with 0.1 µM carvedilol, IR-CAR 1– hearts from IR injury model treated with 1 µM carvedilol, IR-CAR 10– hearts from IR injury model treated with 10 µM carvedilol, IR-MET 0.01 – hearts from IR injury model treated with 0.01 µM metoprolol, IR-MET 0.1 – hearts from IR injury model treated with 0.1 µM metoprolol, IR-MET 1 – hearts from IR injury model treated with 1 µM metoprolol, IR-NEB 0.005 – hearts from IR injury model treated with 0.005 µM nebivolol, IR-NEB 0.05 – hearts from IR injury model treated with 0.05 µM nebivolol, IR-NEB 0.5 – hearts from IR injury model treated with 0.5 µM nebivolol * IR vs IR-CAR 0.01, p<0.05, n=6, ANOVA.
Several concentrations of carvedilol also described in the text, and a single concentration (0.1 micromolar) of this drug was only studied.
Response: We thank the Reviewer #2 for pointing this out. The missing data are now added in Results section which was rewritten according to other suggestions as shown above (Microsoft Word version: page 2, lines 90-95, page 4, lines 135-147, page 7, lines 252-255, PDF version: page 2, lines 88-93, page 4, lines 135-147, page 7, lines 226-229).
This reviewer thinks that this manuscript needs a substantial and very extensive revision before the acceptance for publication.
The scientific English and the presented (or not presented but described) results must be substantially improved and very carefully checked by the Canadian coauthor (CA, Alberta), as a native English speaker, as well.
Response: : We thank the Reviewer #2 for the constructive and insightful comments. We greatly appreciate all suggestions of the Reviewer #2. The scientific English was carefully checked and revised. We have modified the manuscript to address all suggestions given to us which helped us to substantially improve our manuscript.

Round 2
Reviewer 1 Report
The manuscript has been improved by editorial measures. Unfortunately, the authors refrained from doing any of the additional experiments (claiming that time was short). As a consequence, the work is still to preliminary for publication. It raises more question than it answers. The conclusions to be drwan thereof are pure speculation.
I suggest that the authors corroborate their data by additional experimentation elucidating the mechanism behind the phenomea they observe and submit the work de novo,
Author Response
Comments and Suggestions for Authors
The manuscript has been improved by editorial measures. Unfortunately, the authors refrained from doing any of the additional experiments (claiming that time was short). As a consequence, the work is still to preliminary for publication. It raises more question than it answers. The conclusions to be drawn thereof are pure speculation.
I suggest that the authors corroborate their data by additional experimentation elucidating the mechanism behind the phenomena they observe and submit the work de novo,
Response:
We thank the Reviewer #1 for a positive comment of our revision, the possibility to perform additional experiments, and for agreeing to extend the deadline for resubmission. Additional experiments were performed as per the Reviewer #1 suggestion in previous Review Report (Round 1):
It remains unclear, what kind of effects of IR ± CAR on myocardial MMP-2 one is looking at: Induction/repression, release/retention, activation/inhibition of MMP-2? To clarify this crucial question the following data needs to be furnished in addition:
Effect of CAR on MMP-2 activity in vitro or ex vivo.
Response:
As per Reviewer #1 suggestion the effect of carvedilol on MMP-2 activity in vitro was evaluated by developing of zymograms performed on selected heart extracts with addition of carvedilol to the incubation buffer in comparison to control gels (with no drug added), which served as a model of 100% of MMP-2 activity.
The description of the method was added to “ Materials and methods” section ( page 16, lines 358–361), the results of this experiment were added to the “Results” section (page 7, lines 147–149) and a new figure was added to the manuscript (page 8, line 150) with figure legend (page 8, lines 151–155):
“In vitro inhibition of MMPs activity by carvedilol was evaluated by developing of zymograms performed on selected heart extracts with addition of carvedilol to the incubation buffer in comparison to control gels (with no drug added), which served as a model of 100% of MMP-2 activity”.
“2.3. Carvedilol has no effect on MMP-2 activity in vitro. Carvedilol did not inhibited the activity of MMP-2 when run out on gel zymograms incubated with 0.1 μM carvedilol (Fig. 3 a-b).”
“Figure 3. Effect of CAR on MMP-2 activity in vitro. a. Densitometric analysis of gelatinolytic MMP-2 activities in control gel and gel incubated with addition of 0.1 μM CAR. b. Representative zymograms of MMP-2 activity in control gel (A) and gel incubated with addition of 0.1 μM CAR (B). C – control group, C+ CAR 0.1 – in control gel and gel incubated with addition of 0.1 μM CAR. P>0.05, n=8, T-test”
Effect of IR ± CAR on MMP-message (mRNA) in myocardial extracts.
Response:
As per Reviewer #1 suggestion MMP-2 mRNA expression was evaluated in heart tissue. The description of the method was added to “ Materials and methods” section (page 16–17, lines 383–401), the results of the experiment were added to the “Results” section (page 8, lines 156–159) and new figure was added to the manuscript (page 9, line 160) with figure legend (page 9, lines 161–164):
“4.7. Expression of MMP-2 gene in heart tissue
Ribonucleic acid (RNA) was extracted from powdered heart tissue by phe-nol/chloroform technique using PureZol RNA isolation reagent (BioRad), according to manufacturer’s instruction. Briefly, 50 μg of tissue powder was mixed with 1 mL of Pure-Zol, and immediately homogenized by Pellet Pestle® Motor (Kimble Kontes). Next steps included extraction with chloroform (Stanlab), precipitation of RNA with isopropanol (Chempur), and washing with 75% ethanol (Chempur). Purified RNA was dissolved in 50 μL of DEPC-treated water (Ambion) and its quality and concentration were assessed by NanoDrop Lite Spectrophotometer (Thermo Fisher Scientific). 500 ng of RNA was taken for reverse transcription performed using iScript™ cDNA Synthesis Kit (BioRad). Reverse transcription and subsequent real time PCR were both performed on CFX96 Touch Re-al-Time PCR Detection System (BioRad). 100 ng of each cDNA template was used for both genes real-time amplification in duplicates, using iTaq Universal SYBR® Green Supermix (BioRad) following manufacturer’s protocol. Glyceraldehyde 3-phosphate dehydrogenase (GAPDH) served as a housekeeping gene for normalization of MMP-2 gene expression. Primers sequences were designed as follows: GAPDH F: 5’ AGTGCCAGCCTCGTCTCA-TA 3’, GAPDH R: 5’ GATGGTGATGGGTTTCCCGT 3’; MMP-2 F: 5’ AGCAAGTAGAC-GCTGCCTTT 3’, MMP-2 R: 5’ CAGCACCTTTCTTTGGGCAC 3’. Relative fold MMP-2 gene expression was calculated according to delta-delta Ct formula.”
“2.4. Carvedilol does not change MMP-2 mRNA expression in hearts subjected to ischemia-reperfusion. Real time PCR revealed no significant changes in MMP-2 mRNA expression between groups (Fig. 4).”
“ Figure 4. Effect of carvedilol on MMP-2 mRNA expression in hearts subjected to ischemia-reperfusion. C – control group, aerobically perfused hearts, IR injury – ischemia-reperfusion injury, CAR 0.1 – aerobically perfused hearts treated with 0.1 µM carvedilol, IR-CAR 0.1 – hearts from IR injury model treated with 0.1 µM carvedilol. P>0.05, n=5, ANOVA.”
Incorporation of a new paragragh into “Materials and methods” section required the change of the number of next paragraph (page 17, line 402)
Effect of IR ± CAR on MMP-mass (protein) in coronary effluents and myocardial extracts
Response:
As suggested by the Reviewer #1 MMP-2 protein content was evaluated by western blot. It required adding the description of the method to “Materials and methods” section (page 16, lines 362, 373–381). The results of the experiment were added to the “Results” section (page 9, lines 165–170) and new figure was added to the manuscript (page 10, line 171) with figure legend (page 10, line 172–180):
“MMP-2 content was determined in coronary effluents. 30 μL of each concentrated coronary effluent was applied to 10% SDS-PAGE gels at reducing conditions. After electrophoresis samples were electroblotted for 40 min at 50 V onto a nitrocellulose membrane 0.45 μm (BioRad) by wet technique. A primary monoclonal mouse antibody against total MMP-2 ab86607 (Abcam) and secondary goat-anti-mouse conjugated with HRP (BioRad) were both used at a dilution of 1:1000. The blot was developed and scanned as described above. The 72 kDa MMP-2 was detected by comparison with Precision Plus Protein Standards (BioRad) and relative content was calculated on the basis of 75 kDa band intensity and expressed in arbitrary units.”
“ 2.5. Carvedilol does not affect MMP-2 content in coronary effluent. IR injury caused a significant increase in MMP-2 content, assessed by western blot, in coronary effluent in second minute of reperfusion in both IR and IR-CAR 0.1 groups, but there were no significant differences between IR and IR-CAR 0.1 groups (Fig 4. a-b).
There were no significant differences in MMP-2 content in heart tissue between groups (data not shown).”
“ Figure 4. MMP-2 content in coronary effluents and heart tissue. a. Densitometric analysis of MMP-2 content in coronary effluent collected at 2 min of reperfusion after 20 min ischemia in IR CAR 0.1 group as determined by Western blot. b. Representative western blot of MMP-2 content in coronary effluents collected at 2 min of reperfusion after 20 min ischemia in IR CAR 0.1 group.
C – control group, aerobically perfused hearts, IR injury – ischemia-reperfusion injury, CAR 0.1, aerobically perfused hearts treated with 0.1 µM carvedilol, IR-CAR 0.1 – hearts from IR injury model treated with 0.1 µM carvedilol. C vs IR, C vs IR-CAR 0.1, CAR 0.1 vs IR, CAR 0.1 vs IR-CAR 0.1, p<0.05, n=5, ANOVA.”
Incorporating new results required changing of paragraphs’ numbers in “Results” section (pages 10–13, lines 181–208), figures numbers (page 12, line 189; page 15, line 329) and changes in “Statistical analysis” paragraph (page 17, lines 403–409).
All data are expressed as mean +/- SEM. Comparisons between groups were assessed for significance by two-way analysis of variance (ANOVA) or T-test after assessment of normality of distribution. Logarithmic transformation was performed of data which had non-normal distribution to meet requirements of ANOVA analysis. The conclusions drawn after data transformation remain valid for the original data.”
Also, the conclusions were re-written in the “Discussion” section (page 14, lines 268–282):
“The main limitation of our study is that, based on our results, we cannot directly say what is the exact mechanism responsible for the decrease in MMP-2 activity by carvedilol. Based on the results of in vitro experiment we can exclude direct effect of carvedilol on MMP-2 activity. We can also exclude changes at the transcriptional level as we did not observed differences in MMP-2 mRNA expression between groups, indicating that carvedilol regulates MMPs at the post-transcriptional level. Western blot analysis of MMP-2 content in coronary effluent revealed that there were no changes between IR and IR-CAR 0.1 μM group while the increase in MMP-2 activity assessed in zymography, was significantly lower in the IR injury with CAR 0.1 μM group then in IR group.
Moreover, a decrease in MMP-2 activity in the effluent from hearts treated with carvedilol was accompanied by a decrease in tissue troponin I degradation. These results indicate that the changes of MMP-2 activity resulted from its activation in heart tissue and subsequent release into the coronary effluent in the settings of IR but not from changes in enzyme localization. Further research is needed to explain the exact mechanism by which carvedilol inhibits activity of MMP-2.

Reviewer 2 Report
Figure 2: In this Figure "b" is not marked. Only "a", "c", "d" and "e" can be seen. The order must be corrected.
Author Response
Comments and Suggestions for Authors
Figure 2: In this Figure "b" is not marked. Only "a", "c", "d" and "e" can be seen. The order must be corrected.
We would like to thank the Reviewer #2 for their positive assessment of our study and for pointing it out. The order of numbers in figure 2 was corrected (page 5, line 125).
